# Durability and Thermal Behavior of Functional Paints Formulated with Recycled-Glass Hollow Microspheres of Different Size

**DOI:** 10.3390/ma16072678

**Published:** 2023-03-28

**Authors:** Massimo Calovi, Stefano Rossi

**Affiliations:** Department of Industrial Engineering, University of Trento, 38123 Trento, Italy; massimo.calovi@unitn.it

**Keywords:** hollow glass microspheres, recycled glass filler, paint thermal behavior, coating durability, paint mechanical features

## Abstract

This study aims to assess the effect of hollow glass microspheres of different sizes derived from glass industry waste on the durability and thermal behavior of waterborne paint. The coatings were characterized by electron microscopy to investigate the distribution of the spheres and their influence on the layer morphology. The impact of the various glassy spheres on the mechanical feature of the coatings was assessed using the Buchholz hardness test and the Scrub abrasion test. The role of the spheres in altering the durability of the samples was analyzed by the salt spray exposure test and the electrochemical impedance spectroscopy measurements. Finally, a specific accelerated degradation test was carried out to explore the evolution of the thermal behavior of the composite coatings. Ultimately, this work revealed the pros and cons of using hollow glass spheres as a multifunctional paint filler, highlighting the size of the spheres as a key parameter. For example, spheres with adequate size (25–44 µm), totally embedded in the polymeric matrix, are able to reduce the thermal conductivity of the coating avoiding local heat accumulation phenomena.

## 1. Introduction

Nowadays, hollow glass microspheres represent a crucial component of lightweight multifunctional composite materials and are frequently used in aerospace [1], ships [2,3], flame retardant [4,5], dielectric materials [6,7], and hydrogen storage [8,9], thanks to their lightweight, low permittivity and low thermal conductivity [4,10]. Their low conductivity makes these materials highly functional in thermal insulation applications [5,11,12,13]. From this point of view, glass microspheres can be easily applied as thermal insulation materials for energy-saving buildings [14].

Continuous requests for improved life comfort have become the cause of global warming [15]. The high consumption of fossil fuels like coal, fuel, and natural gas has created unsustainable issues in our society’s use of energy [16]. Today, 20–30% of the world’s energy demand is employed to power air conditioners and other refrigeration equipment [17]. Moreover, In the EU, buildings account for 36% of CO_2_ emissions and 40% of energy usage [18]. The so-called urban heat island effect (UHI effect) has risen due to the growing population in cities and the expansion of metropolitan regions [19,20]. Current research demonstrates that for every degree increase in ambient temperature, the peak electricity demand rises by 0.45% to 4.6%. It translates to a penalty of around 21 ± 10.4 W per person for each degree of temperature increase [21]. Thus, reducing the energy consumption of refrigeration equipment while guaranteeing good summertime living comfort is crucial, also because of the dual problem of the actual energy shortage and a heat island effect. In addition to making substantial use of refrigeration technology, the construction aspects currently require the most energy. According to statistics, the energy consumption by a building’s structure due to heat transmission makes up roughly 25% of that building’s overall energy use [22]. Hence, increasing the thermal insulation of building materials is a key strategy to lower a structure’s energy consumption.

Conventional exterior wall thermal insulation materials have the drawbacks of being expensive, bulky, and having poor heat insulation properties [23]. Conversely, hollow glass microspheres benefit from low cost, low density, superior mechanical attributes, and good fluidity [24,25]. Furthermore, the inclusion of these materials in composite structures has been shown to reduce their heat conductivity [26,27] and flammability [28,29]. Moreover, several studies have investigated the thermal insulation impact of the glass microspheres applied as fillers in polymeric matrices [5,30,31,32,33,34]. Therefore, hollow glass microspheres appear to be excellent candidates as paint fillers for insulating structures.

However, the glass manufacturing sector also significantly impacts the global economy [35]. The previous linear unsustainable economy contributed to rising raw material costs, resource depletion, irreversible environmental degradation, and waste accumulation [36]. Hence, a new regenerative economy is emerging to reduce the environmental impact, built on crucial components like the circular production/consumption system [37]. In this context, the production of microspheres from glass waste are crucial aspects of the large-scale application of this type of material. Nevertheless, glass recycling is still a growing industry with several limitations. For example, according to some studies, several wealthy countries have increased the size of their expected landfills to fit around 200 million tons of glass garbage annually, with a very poor recycling rate [38].

Consequently, this study wants to solicit the use of microspheres deriving from glass recycling processes. This work aims to evaluate the impact of hollow glass microspheres produced by glass waste on the thermal behavior of water-borne paint. Moreover, the various literature works investigating the applications of glass microspheres mainly focus on the impact of the filler on the mechanical and insulating properties of the composite without considering the effect of the microspheres on the durability of the coating. In fact, in some cases, the addition of filler leads to a reduction in the durability of the paint due to inhomogeneity in the bulk of the layer [39], agglomeration phenomena [40], decay of the filler [41] or hydrophilicity features of the additive [42,43]. Thus, this study pays particular attention to this aspect, exploring the influence of the addition of glassy filler on the protective features of the polymeric matrix of the paint, considering the effect of microsphere size as a key parameter.

For this reason, three commercial hollow glass microspheres possessing different dimensions were employed in the study. The glassy fillers were provided by Chimiglass (Riccione, RN, Italy) and Poraver GmbH (Schlusselfeld, Germany). They were produced from recycled glass, binder, and expansion agent, thus possessing highly environmentally friendly features.

The morphology of the composite coatings was analyzed using an optical microscope and scanning electron microscope (SEM) observations. The effect of the glass microspheres on the mechanical characteristics of the layers was assessed by employing the Buchholz Hardness Indentation test and the Scrub test. Moreover, salt spray chamber exposure and Electrochemical Impedance Spectroscopy (EIS) measurements were carried out to examine the protective behavior of the coatings and evaluate the contribution of the recycled filler. Finally, the durability of the coatings was assessed by subjecting the samples to a particular accelerated degradation test, studying the evolution of the thermal behavior of the composite layers, and analyzing the role of the glassy spheres.

## 2. Materials and Methods

### 2.1. Materials

The glass hollow spheres called C40 and C15 were provided by Chimiglass (Riccione, RN, Italy), while the product poraSpheres (called P) were supplied by Poraver GmbH (Schlusselfeld, Germany). Acetone was purchased from Sigma-Aldrich (St. Louis, MO, USA) and used as received. The carbon steel substrate (Q-panel type R (0.15 wt.% C-Fe bal.) −40 mm × 70 mm × 2 mm dimensions) was provided by Q-lab (Westlake, OH, USA). The acrylic-based white primer paint ECOFILLER EQW and the polyurethane-acrylate transparent top-coat paint IDROPUR ZW 01 were supplied by EP Vernici (Solarolo, RA, Italy).

### 2.2. Samples Production

The metallic substrates were appropriately pre-treated before painting to increase the coating’s adherence. First, two minutes of ultrasound treatment in acetone were used to degrease the metal plates. Next, the mechanical pickling was completed using a sandblasting procedure with alumina particles (0.2 mm diameter-70 mesh), providing the steel substrates with a roughness [Ra] of 3.05 ± 0.21 µm. Lastly, a second acetone degreasing step was carried out to eliminate potential contamination residues.

The spray application method was employed to obtain a primer layer with a thickness of around 110 µm. Next, the samples were cured in an oven at 60 °C for 30 min before the subsequent top-coat application. Therefore, the polyurethane-acrylate clear top-coat paint was applied in two steps and allowed to dry in room T for fifteen minutes between each. Finally, to complete the curing of the topcoat, the samples were heated in an oven at 60 °C for 60 min. As a result, the final coatings, which included the primer and the topcoat, exhibited a thickness of around 210 µm.

Three kinds of hollow glass spheres (1 wt.%) were added to the top-coat paint mixture, and the three formulations were mixed for 30 min using an ultrasound probe to aid in the homogenous dispersion of the microspheres. The performance of the polyurethane-acrylate top-coat layer, free of the recycled glass spheres, was compared with the performance of the three series of glass microsphere coatings. Table 1 summarizes the samples with their nomenclature.

### 2.3. Characterization

To investigate the affinity between the polymeric matrix and recycled filler, the morphology of the microspheres and the cross sections of the coatings were observed using the low vacuum scanning electron microscope SEM JEOL IT 300 (JEOL, Akishima, Tokyo, Japan). The SEM pictures were examined employing ImageJ software (version 1.53t), to analyse the size distribution of the spheres. The roughness of the coatings was investigated with the surface roughness measurement instrument MarSurf PS1 (Carl Mahr Holding GmbH, Gottingen, Germany).

Using both the Buchholz Hardness Indentation test and the Scrub test, the impact of the glass microspheres on the mechanical characteristics of the polyurethane-acrylate topcoat was assessed. During the Buchholz test, the length of the indentation created by the standardized instrument was measured, following the UNI EN ISO 2815 standard [44]. The Scrub test was carried out employing an Elcometer 1720 Abrasion and Washability Tester, following the BS EN ISO 11998 standard [45]. The samples were subjected to 1000 scrub cycles (37 cycles per minute) in dry mode. The weight loss of the coatings was evaluated every 250 cycles.

Accelerated degradation tests were used to examine the protective behavior of the coatings and evaluate the contribution of the recycled filler. To determine how the microspheres would affect the protective performance of the composite layers in a specific hostile environment, the samples were placed in a salt spray chamber (Ascott Analytical Equipment Limited, Tamworth, UK) for 500 h, according to the ASTM B117-11 standard [46] (5 wt.% sodium chloride solution). An artificial cut was realized on the surface of the samples to assess the adhesion of the coatings, evaluating possible phenomena of coating detachment and water uptake. Moreover, the protective features of the composite coatings were investigated by means of Electrochemical Impedance Spectroscopy (EIS) measurements, carried out with a potentiostat Parstat 2273 (Princeton Applied Research by AMETEK, Oak Ridge, TN, USA) with the software PowerSuit ZSimpWin (version 2.40) and applying a signal of about 15 mV (peak-to-peak) amplitude in the 10^5^–10^−2^ Hz frequency range. The cell setup comprises an Ag/AgCl reference electrode (+207 mV SHE) and a platinum counter electrode immersed in the 3.5 wt.% sodium chloride aqueous solution. The samples were immersed in the test solution for 500 h, with a testing area equal to 6.5 cm^2^. The measurements were carried out on five samples per series.

Finally, the impact of the microspheres on the thermal behavior of the paint was evaluated employing the experimental setup depicted in Figure 1 and already optimized in previous works [47,48]. The thermal behavior of the paint was evaluated considering the thermal conductivity of the coating and heat transmission within the house model represented in Figure 1, measuring temperature changes in the system. The measurement setup consisted of a 150 × 150 × 2 mm^3^ coated sample located as roof panels on a roofless box (200 × 270 × 200 mm^3^) made of polyurethane foam sheets. Each sample was subjected to a 150 W IR-emitting lamp (Philips IR150R R125, Eindhoven, The Netherlands) located at a distance of 260 mm. Two thermocouples PT 100 (temperature sensors) were employed to collect the temperature data: the first placed on the rear part of the coating panel to evaluate the panel temperature T_surf_, and the other in the middle of the box, at 100 mm from the coated panel, to measure the small-scale house internal temperature T_int_. The thermocouples were connected to a Delta OHM HD 32.7 RTD (Delta OHM Srl, Selvazzano Dentro, Italy) data logging instrument for a temperature data recorder every 5 s. DeltaLog 9 software was employed to control the instrument. For each measurement, the temperature was monitored for 27 min, during which the IR lamps remained on, plus an additional 30 min to allow the samples to return to room temperature.

Each thermal measurement was interspersed with an accelerated degradation cycle according to the ISO 20340:2009 standard [49], consisting of the following:72 h of exposure to UV and water in accordance with ISO 11507:1997 [50], alternating periods of 4 h exposure to UV-A (340 nm) at (60 ± 3) °C and 4 h exposure to condensation at (50 ± 3) °C, employing a UV173 Box Co.Fo.Me.Gra (Co.Fo.Me.Gra, Milan, Italy);72 h of exposure to salt spray in accordance with ISO 7253:1996 [51], using a salt spray chamber (Ascott Analytical Equipment Limited, Tamworth, UK);24 h at room temperature.

After each degradation cycle, the samples were characterized by measuring their thermal performance and appearance properties. In addition, the coatings’ possible degradation was monitored by FTIR infrared spectroscopy measurements and colorimetric analyses. The FTIR spectra were acquired with a Varian 4100 FTIR Excalibur spectrometer (Varian Inc., Santa Clara, CA, USA) to investigate the chemical modifications of the polymeric matrix. The colorimetric measurements were carried out by means of a Konica Minolta CM-2600d spectrophotometer (Konica Minolta, Tokyo, Japan) with a D65/10° illuminant/observer configuration in SCI mode.

## 3. Results and Discussion

### 3.1. Powders and Coatings Morphology

Figure 2 shows the morphology and size distribution of the three different microspheres. The products C40 and C15 (Figure 2a,b, respectively) possess similar characteristics in terms of high circularity and thickness of about 1µm (analyzed with SEM). While the powders C40 are particularly small, with an average diameter of about 25 µm, the dimensional range of the microsphere C15 is wider, with an average value of about 44 µm. Otherwise, the filler P (Figure 2c) reveals very different characteristics. The circularity of the spheres is not constant, and the thickness of the walls can even exceed 5 µm.

Furthermore, their surface is not as smooth as that of the previous two powders but rather rough. Finally, there is a clear dimensional difference between the powders C40 and C15: the spheres P have an average size of about 130 µm, but in rare cases, they can even exceed 200 µm in diameter. This is an aspect of particular importance, as the performance of the filler can vary according to the dimensional relationship with the layer in which they are added.

Thus, the morphology of the coatings containing the recycled spheres was deeply investigated. Figure 3 reveals the four samples’ appearance in the top view (on the left) and the cross-section (on the right). The top-view micrographs were acquired with the optical microscope, while the sections of the coatings obtained by brittle nitrogen fracture were observed by SEM. The top-view images show a change in the surface morphology of the coating according to the type of microsphere. The very small powders C40 and C15 are homogeneously distributed in the coating and can also be observed on the surface of the layer. This aspect is more evident for the spheres P, whose large size makes them easily distinguishable within the transparent topcoat. Since the three types of filler have been added in the same percentage in the topcoat, the larger and heavier P powders appear sparser in the coating.

The cross-sections of the samples show the spheres tightly adhered within the polymeric matrix. During the brittle fracture process in nitrogen, the microspheres undergo breakage and sectioning but are not expelled from the top-coat bulk. This phenomenon is representative of the positive affinity between polymer and filler. The compatibility between the fillers and paint matrix concerns physical and geometric aspects rather than the adhesion between the polymer and glass. The absence of voids and spaces at the polymer-glass interface facilitates these compatibility issues, although they don’t necessarily refer to good adhesion. The spherical glass filler is well received and surrounded by the polymer matrix. The fracture of the coating does not result in the release of the entire sphere but rather in its fracture into multiple components. The spheres C40 and C15 possess dimensions compatible with the top-coat thickness: most of the powders are completely immersed in the polymeric matrix, but many sprout slightly on the surface of the coating. Otherwise, the powders P are so large that they partially escape from the polyurethane-acrylate layer. At the same time, their presence leads to a modification of the morphology of the coating, which tries to accommodate the individual powders, with a consequent section of the layer that is irregular and full of depressions and prominences. This phenomenon is highlighted in Figure 4, representing the jagged surface morphology of coating G3. These bumps are made of coated P spheres, too large to be completely incorporated into the top-coat thickness.

Consequently, the presence of the three types of spheres leads to a non-negligible increase in the surface roughness of the composite coating. Table 2 summarizes the values of surface roughness [Ra] of the coatings, as the average value of 50 measurements carried out on 10 samples per series (5 measurements per single sample). The increase in roughness is strictly connected to the presence and size of the microspheres: the clear difference in diameter between the powders C15 and P results in a completely different level of roughness between the samples G2 and G3.

Ultimately, adding the microspheres involve a significant morphological modification of the coating, both regarding its internal structure and surface features. The size of the spheres is a determining factor: too large fillers cause an increase in the roughness of the coating, as they are not completely covered by the polymeric matrix. However, all three microspheres demonstrated good compatibility with the polymer matrix, remaining well incorporated into the composite layer despite destructive stresses such as those introduced by the brittle fracture process. Consequently, the microspheres are candidates as paint fillers, provided that the dimensional relationship between the diameter of the sphere and the thickness of the coating is considered.

### 3.2. Effect of the Glass Spheres on Paint Mechanical Features

#### 3.2.1. Buchholz Hardness

The Buchholz Hardness Indentation test was employed to assess the influence of the recycled filler on the hardness of the paint. Figure 5 refers to the average length of the indentations produced by the Buchholz disc indenting tool, with the corresponding Buchholz hardness value. The result is the outcome of 30 measurements, as 10 measurements were carried out on 3 different samples for each series of samples (30 measurements in total). The gauge was placed onto the coating for 30 s. A graduated microscope was used to measure the length of any consequent indentations in the coating. The units of Buchholz Indentation Resistance were stated using the scale specified by the UNI EN ISO 2815 standard [44].

Although the walls of the microspheres are very thin, they exhibit a greater hardness than the pure polymeric matrix, which is particularly soft, as their addition leads to a slight reduction in the length of the indentation (samples G1 and G2). The filler is so small that it can offer good resistance to the indenter; however, the final Buchholz hardness value of the coating is still very low (<50). The standard deviation of the mean values of samples G1 and G2 is quite significant, a symptom of a certain inhomogeneity in the acquired results. Conversely, the larger powders of sample G3 do not show such a significant change in hardness compared to the filler-free coating G0. In this case, as the microspheres are more scattered on the surface of the coating, as shown in Figure 3d, their contribution is less significant, while the soft polymeric matrix is predominant. As a result, the hardness values of samples G0 and G3 are very similar. Also, in this aspect, the size of the filler plays a key role in influencing different aspects of the composite coating.

#### 3.2.2. Scrub Test

The graph in Figure 6 highlights the results of the Scrub test. The loss of mass due to the repetitive movement of the abrasive sponge appears to increase due to the presence of the microspheres. The size of the filler also influences this phenomenon: the larger the microspheres, the greater the mass loss per unit area measured during the test. For example, after 1000 cycles, the mass loss of sample G3 is nearly double that of coating G0, free of filler. Otherwise, the behavior of sample G1 is almost comparable with that of the pure polymeric matrix.

The reason for these results is portrayed in the images in Figure 7, acquired by SEM after 1000 Scrub cycles. The surfaces of the four coatings represent the typical abrasion lines caused by the Scrub abrasive sponge [52]. The cyclic sliding movement of the sponge causes the development of shear stresses on the surface of the composite layer. These shear stresses result in the fracture of the glass microspheres located in the proximity of the surface. Consequently, the mass loss measured during the Scrub test is associated with the structural decay of the polymer matrix and with the breakage of the filler, which the abrasive sponge removes. This phenomenon is more important the more the microspheres emerge from the bulk of the layer and are subject to shear stresses.

Furthermore, for the same number of spheres destroyed, the loss of mass will be greater as a function of the size of the filler. Both factors result in sample G3 showing the greatest mass loss during the test. The microspheres P are the largest (greatest mass) and consequently are less protected by the coating and undergo greater shear stresses. The asperities shown in Figure 4 are more subject to abrasion phenomena and, therefore, are more easily removed from the abrasive sponge. The images in Figure 7 highlight several circular dark holes on the surfaces of the coatings containing the glass fillers. These events represent the microspheres, literally sheared in two parts by the sponge: the upper part of the spheres has been removed, while the lower part is still well adhered to and immersed in the polymeric matrix. However, some microspheres remain intact: the powders are most covered and protected by the polymeric matrix. The smaller the spheres, the more likely this event will occur.

Consequently, on the surface of sample G1, it is still possible to appreciate many intact spheres. This results in an almost negligible increase in mass loss with respect to the polymeric matrix (sample G0). The number of intact fillers decreases in sample G2, disappearing almost completely in coating G3.

Ultimately, the size of the microspheres also represents a key factor in the mechanical performance of the coating. The powders, if small enough to be completely incorporated in the polymeric matrix, can slightly reduce the indentation phenomena, increasing the overall hardness of the coating. Otherwise, the most exposed spheres are easily destroyed by the Buchholz indenter and the abrasive sponge of the Scrub test. The recycled fillers are unable to influence the mechanical features of the coating significantly but are easily degraded if exposed to mechanical stress. Spheres with a diameter greater than the thickness of the polymeric layer are destined for considerable physical degradation. Consequently, this type of glassy microsphere must be adequately protected by the polymeric matrix.

### 3.3. Effect of the Glass Spheres on Paint Durability

The recycled filler can significantly alter the morphology of the coating. This aspect is strictly connected to the protective performance of the composite layer. Therefore, the samples were subjected to an accelerated degradation test and electrochemical measurements to evaluate the impact of the glass microspheres on the durability of the coating.

#### 3.3.1. Salt Spray Test

Figure 8 shows the evolution of the appearance of the samples during exposure to the salt spray chamber. Sample G0 reveals excellent protective properties, as no phenomena of water uptake and development of blisters are observed, not even in proximity to the artificial notch. Recent works [40,53] highlight some protective gaps of waterborne paints, which easily absorb test solutions and exhibit several blisters. Otherwise, the thickness of coating G0 in this work is so high as to guarantee a good barrier effect against the penetration of water and aggressive ions. However, the same behavior was also observed in the other three samples. The presence of glass microspheres modifies the surface morphology of the topcoat but does not favor the absorption of test solutions inside the polymeric matrix. The compatibility between microspheres and polymer is so good that the interface between filler and matrix does not represent a discontinuity capable of allowing water penetration and aggressive substances. Consequently, none of the samples shows blisters on their surface, not even after 500 h of salt spray test exposure.

At the end of the exposure test, to better observe the damage caused by the artificial notch, the samples were immersed in a 5 g/L solution of citric acid for 10 min with ultrasound treatment to remove any corrosion product. After this process, it was possible to evaluate the extent of the cathodic delamination phenomenon and the detachment of the coatings near the notch. The appearance of the samples is highlighted on the right in Figure 8 (post 500 h) after adequate removal of the layer, which has lost adherence to the metal substrate employing a scalpel. Table 3 expresses the values of the distance of the coating detachment for the notch, used as a parameter for the adhesion evaluation of the composite layers. The values were obtained by observing the samples with the optical microscope and measuring the length of the detachment of the coating from the central notch. The standard [46] establishes 1000 µm as the maximum detachment distance value of the coating from the scratched area. Therefore, the four series of samples do not possess optimal adhesion values, as they exhibit detachment distances greater than 3000 µm. However, the glass microspheres do not negatively affect this aspect either, as the results of the four samples are comparable.

In conclusion, the filler-free coating possesses intrinsically good protective properties and interesting durability, even if its adhesion could be improved by adequate pre-treatments. Despite the important morphological changes introduced by the glass microspheres, the recycled filler does not seem to affect the performance of the coating negatively. However, the microspheres do not promote the absorption of test solutions and consequently do not favour cathodic delamination phenomena, reducing the coating adhesion.

#### 3.3.2. Electrochemical Impedance Spectroscopy Measurements

Electrochemical Impedance Spectroscopy (EIS) measurements are often used to evaluate the corrosion resistance properties of organic coatings [42,43], analyzing their defects or degree of adhesion [41]. The Bode impedance moduli measured at low frequencies (10^−2^ Hz), referred to as |Z|_(0.01)_, is a variable that offers a rough quantitative estimate of the level of protection supplied by the organic layer. In this context, some literature works [54,55] designate the coatings as ‘protective’ if they possess a |Z|_(0.01)_ value greater than 10^6^ Ω cm^2^.

Consequently, the four series of coatings’ protective behavior was determined by recording the development of their impedance module |Z|_(0.01)_ through time. Figure 9 depicts the parameter |Z|_(0.01) changes_ throughout the samples’ 500-h contact with the test solution. The test result reveals that the coatings of the four series offer appropriate protection to the steel substrate since all four samples exhibit |Z|_(0.01)_ values that never drop below the threshold limit of 10^6^ Ω cm^2^. This outcome agrees with the results of salt spray chamber exposure, as all four types of coatings show good protective behavior. However, the EIS measurements offer more detailed information: sample G0 shows a trend of |Z|_(0.01)_ values approximately one order of magnitude greater than the three coatings containing the glassy microspheres. The filler causes the decrease of |Z|_(0.01)_ as the fill introduces a structural discontinuity in the bulk of the coating, which is ‘felt’ as a defect of the electrochemical system. This phenomenon was not appreciable from the salt spray test, but it appears evident from the evolution of |Z|_(0.01)_. Regardless of this aspect, the four types of samples show a rapid collapse of the impedance modulus during the first hours of testing, which then slowly increases over time. This phenomenon, already observed in previous studies [50,56], represents slight absorption processes of test solution inside the coating, which increases the capacitive contribution of the system.

The development of the Bode phase spectra is represented in Figure 10 to critically evaluate the four samples’ behavior and explain the findings in Figure 9. At the beginning of the test (*t* = 0 h), sample G0 shows a single asymmetric broad peak corresponding to the high frequency time constant, which is related to the dissipation events that occur through the coating [57,58]. However, the phase angle values are relatively high in a wide range of frequencies, representing a good capacitive and protective contribution of the coating. Thus, during the first 24 h, the spectrum of sample G0 exhibits a shift to higher frequencies as a symptom of a reduction in the insulating properties. Subsequently, the spectra show a typical tendency: the continuous shift of the curves towards lower frequencies due to an increase in the capacitive contribution of the system due to the continuous absorption of the test solution. This event confirms the increase in |Z|_(0.01) value_ over time observed in Figure 9.

The behavior and trend of the three samples containing the recycled filler appear very similar: a large shift in the curve at high frequencies during the first 24 h, followed by a continuous movement towards lower frequencies. Apart from the spectrum at *t* = 0 h, the curves of the coatings with the microspheres are superimposable at the same time, as the three coatings provide the same protective behavior. Ultimately, compared with the output of sample G0, the glass spheres slightly alter the capacitive behavior of the coatings but do not significantly inhibit the protective performance of the polymeric matrix.

In conclusion, the accelerated degradation test in an aggressive environment does not allow for highlighting the true impact of the glass microspheres on the protective properties of the coatings. Only through EIS measurements is it possible to underline a slight difference in the behavior of the samples due to the presence of fillers. However, the recycled glass powders do not substantially reduce the coating’s protective features, regardless of their size. The adequate affinity between the glassy spheres and polymeric matrix allows these micropowders to be added to coatings and paints without negatively affecting their durability.

#### 3.3.3. Accelerated Degradation Test

The accelerated degradation cycles were interspersed with FTIR measurements to assess the effect of the microspheres on the integrity of the polyurethane-acrylate matrix. Figure 11 shows the evolution of the FTIR spectra of the samples before and after the 6 cycles of the accelerated degradation test. Before the degradation test (cycle 0), sample G0, free of filler, exhibits a spectrum with an absorption band between 3400 and 3300 cm^−1^, representing the characteristic urethane and urea groups stretching vibration of the NH bond. Moreover, the stretching region between 3000 to 2800 cm^−1^ represents the −CH and −CH_2_ groups. The two intense peaks at about 1718 cm^−1^ and 1681 cm^−1^ refer to the carbonyl absorption band and the urethane and urea carbonyl groups, respectively [59]. The band at 1529 cm^−1^ is attributed to the N-H deformations [60], while the signals at 1460 cm^−1^ and 1375 cm^−1^ can be assigned to the bending of CH_2_ aliphatic. The peaks at 1240 cm^−1^ and 1138 cm^−1^ correspond to N-H bending and the coupled C-N and C-O stretching vibrations. Finally, the peaks at 844 cm^−1^ and 764 cm^−1^ are associated with the C-H stretching and the ester C-O-C symmetric stretching vibration, respectively [61].

The exposure of sample G0 to the degradation cycles does not seem to have caused any chemical-physical degradation of the polymeric matrix of the coating since the spectrum of cycle 6 is perfectly superimposable on the initial one. The first of the three stages of any degradation cycle consist of exposing the samples to UV radiation, which usually leads to the disruption of the chemical structure of a polymer, resulting in a drop in the polymer’s molecular weight and a loss of its mechanical properties. “Photo-oxidative degradation” is the term used to describe this process [62]. Nonetheless, it has been demonstrated that waterborne polyurethane-acrylate coatings possess good resistance to chemical-physical deterioration provoked by ultraviolet light [63]. This aspect explains why the FTIR spectra of sample G0 remain unchanged following the 6 cycles of the degradation test. Moreover, the presence of the glass filler is not appreciable by the FTIR spectra, nor does it seem to influence the durability of the polymeric matrix. The spectra of samples G1, G2 and G3 are identical to those of coating G0 and do not vary following the degradation test.

The results of the colorimetric investigations, however, were substantially different. Figure 12 reveals the evolution of the color ΔE of the samples as a function of the degradation cycles. The ASTM E308 (2018) standard [64] is followed to compute the total color variation ΔE, following the equation:ΔE = [(ΔL*)^2^ + (Δa*)^2^ + (Δb*)^2^]^1/2^(1)
where L*, a*, and b* are the colorimetric coordinates for lightness (0 for black and 100 for white), the red-green coordinate (positive values are red, negative values are green), and the yellow-blue coordinate (positive values are yellow, negative values are blue), respectively.

The graphs in Figure 12 show a clear color change in all samples due to the first cycle of degradation. The change in the appearance of the four samples can be considered substantial because the literature classifies a color change ΔE equal to 1 as discernible even to the human eye [65]. Consequently, the degradation test, consisting of exposure to UV radiation and a salt spray chamber, involves an effective degradation of the coatings, highlighted by a clear and immediate color change, similar for all samples. The color of the various coatings does not undergo further particular variations following the first cycle, suggesting a rapid and complete chemical-physical degradation of the surface of the layers. However, the ΔE appears less significant as the size of the microspheres increases. The greater the surface covered with glass filler, the less the polymeric matrix is exposed to degradation; consequently, the accelerated test’s overall color change is less affected. At the end of the test, sample G3 shows a color change 3 points lower than that of the pure polymeric matrix (sample G0).

The impact of the microspheres on paint performance is also evident from the results of the thermal measurements. Figure 13 represents the evolution of the maximum value of T_surf_ and T_int_ measured with the setup described in Figure 1 when the system stabilizes, and the temperature reaches a plateau value. The temperature values tend to increase with the degradation cycles due to the decay of the polymeric matrix. This temperature increase is more marked near the sample surface (T_surf_), while it grows more slowly inside the lab-scale house (T_int_). The presence of smaller size microspheres (samples G1 and G2) results in lower measured temperature values. The glass fillers create air gaps in the bulk of the coating [66]. Since the thermal conductivity of the air is approximately 10 times lower than that of the polyurethane-acrylic polymers [67], the presence of the microspheres reduces the coating’s ability to transmit heat. The impact of the filler is greatest on the temperature measured near the sample, but it also affects and reduces the values of T_int_.

However, this phenomenon is not respected by sample G3, containing the spheres P. The huge size of the filler P introduces larger air voids into the coating, thus suggesting better insulating power. Furthermore, the lower surface degradation shown in Figure 12 proposes a better protective behavior of the larger spheres. However, the contribution of the microspheres P is not as functional as that of the fillers C40 and C15. The temperature values measured with sample G3 are similar to those of the pure polymer matrix (G0 coating). The hypothesis for the heat transfer mode through a polymer/hollow glass microspheres composite coating has been proposed in a recent study [68]: the fillers, homogeneously distributed inside the coating, realize the so-called ‘thermal insulated islands’, which can block the heat transfer and prolong the heat transfer path across the polymer matrix. With the same weight content (1 wt.%), the coating contains a higher number of spheres C40 and C15 than the one P. The smaller-sized microspheres can, therefore, limit the heat flow through the coating. The top-view image in Figure 3d reveals large filler-free coating spaces in which the heat flow is not counteracted by the large glass microspheres P. Consequently, the filler P slows down the heat flow only in localized areas but is not able to significantly increase the thermal insulation properties of the entire system.

Furthermore, the sphere P produces two local overheating phenomena, similar to the greenhouse effect [69], schematically shown in Figure 14. The large spheres P, partially uncovered by the coating, act as lenses, reflecting a portion of the infrared radiation and focusing the thermal emission in localized areas. Figure 14a shows the phenomenon of an intact sphere, with a consequent local increase in the surface temperature of the coating near the filler. However, some spheres on the surface of the coating can be damaged, causing the radiation to reflect within it (Figure 14b). This process could lead to a local increase in temperature within the coating along the surface of the sphere. Both phenomena cause the reduction in performance observed in the graph in Figure 13.

Otherwise, at the end of the degradation test (cycle 6), the microspheres C40 and C15 lead to a reduction of Tsurf equal to about 3.5 °C and 3.0 °C, respectively, compared to the pure polymeric matrix. Moreover, the temperature Tsurf of the two samples, G1 and G2, after 5 cycles is even lower than that of sample G0 before the accelerated degradation test starts.

Also, in this case, the size of the filler plays a key role in influencing the performance of the polyurethane-acrylic coating. The greater the voids within the coating, the greater the insulating contribution of the microspheres. However, the microspheres must be homogeneously dispersed in the coating to increase the heat transfer path across the polymer matrix to obtain a significant insulating contribution. In this view, very large fillers, such as the spheres P, are non-functional, even though they introduce large voids within the bulk of the coating. Moreover, the fill must be completely incorporated in the bulk of the coating to avoid undesirable phenomena of local heat accumulation, which cause an increase in the temperature of the coating. Fillers of adequate size, on the other hand, can reduce the thermal conductivity of the coating, as in the case of samples G1 and G2.

## 4. Conclusions

The effect of the size of recycled glass hollow microspheres on the durability and thermal characteristics of polyurethane-acrylate paint has been evaluated in this work. The size of the microspheres represents a crucial factor, as it influences the protective and thermal contribution of the filler, modifying the morphology of the composite coating. In addition, very large spheres introduce various voids into the polymeric layer and increase its surface roughness.

The change in morphology is strictly correlated to the coatings’ mechanical features: the roughness introduced into the coating causes a reduction in resistance to abrasion, as the glassy powders are easily destroyed by the shear stress to which they are subjected. Otherwise, the powders, if small enough to be completely incorporated in the polymeric matrix, can slightly reduce possible indentation phenomena, increasing the overall hardness of the coating. Consequently, this type of glassy microsphere needs to be adequately protected by the polymeric matrix to provide a beneficial mechanical contribution.

Differently, recycled glass powders do not substantially reduce the coating protective features, regardless of their size. The good compatibility between the filler and polymeric matrix allows these micropowders to be added to coatings and paints without negatively affecting their durability, as confirmed by the exposure in the salt spray chamber and the EIS measurements.

Finally, the physicochemical decay of the coatings produced by the accelerated degradation test results in the reduction of the insulating power of the composite paint. However, fillers of adequate size can reduce the thermal conductivity of the coating, as in the case of samples G1 and G2. Thus, the microspheres must be completely incorporated in the bulk of the coating to avoid undesirable phenomena of local heat accumulation, which cause an increase in the temperature of the coating.

In conclusion, this work highlights the positive and negative effects of glass microspheres made from waste and recycled material, the size of which plays a key role in the performance of the filler. Microspheres of the appropriate size for the polymeric layer can improve its mechanical properties but increase the insulating features of the coating. Therefore, these coatings, in addition to employing recycled materials, appear interesting for structural applications in which it is necessary to reduce heat absorption by the building without additional electricity.

## Figures and Tables

**Figure 1 materials-16-02678-f001:**
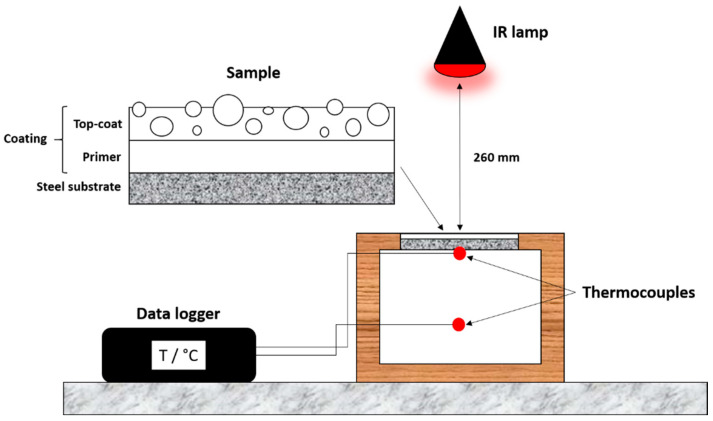
Experimental set-up used for the thermal behavior measurements.

**Figure 2 materials-16-02678-f002:**
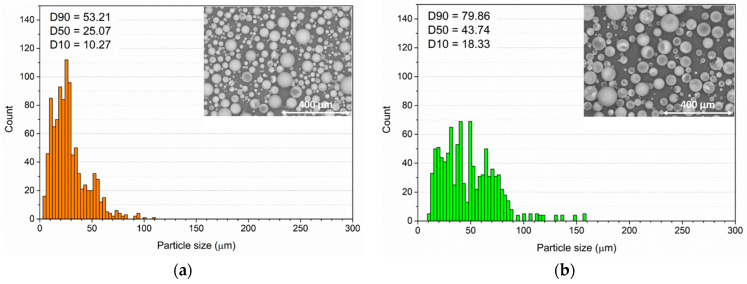
Evaluation of the glassy spheres’ dimensions carried out with ImageJ on (**a**) spheres C40, (**b**) spheres C15 and (**c**) spheres P. The pictures of the spheres were acquired by SEM (scale bar: 400 µm).

**Figure 3 materials-16-02678-f003:**
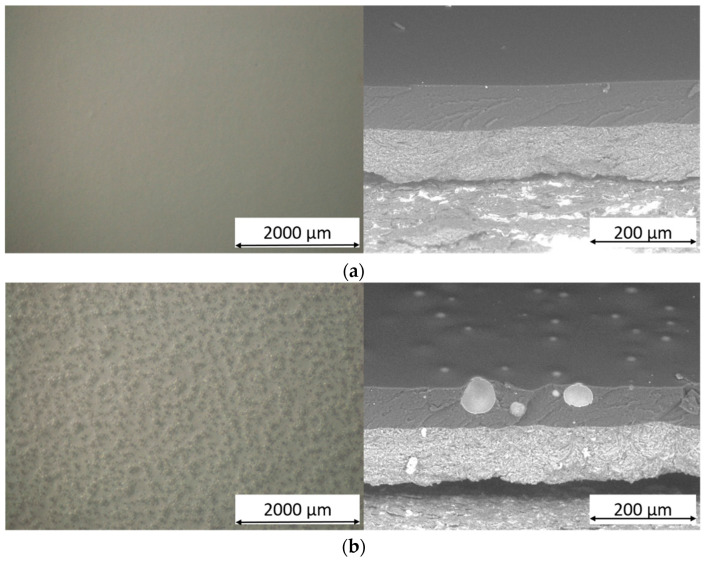
Optical microscope micrographs of the top-view (on the left) and SEM micrographs of the cross-section (on the right) of (**a**) sample G0, (**b**) sample G1, (**c**) sample G2 and (**d**) sample G3.

**Figure 4 materials-16-02678-f004:**
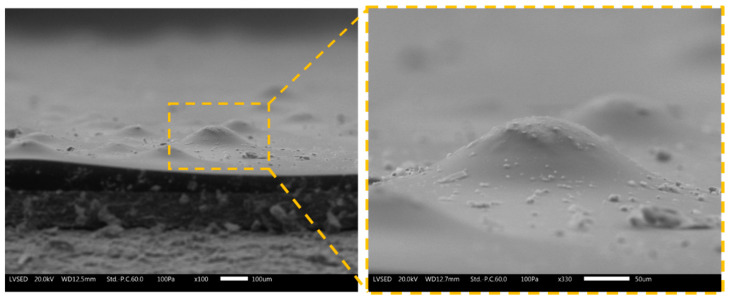
SEM micrograph of the cross-section of coating G3, focusing on the surface roughness due to the glass spheres.

**Figure 5 materials-16-02678-f005:**
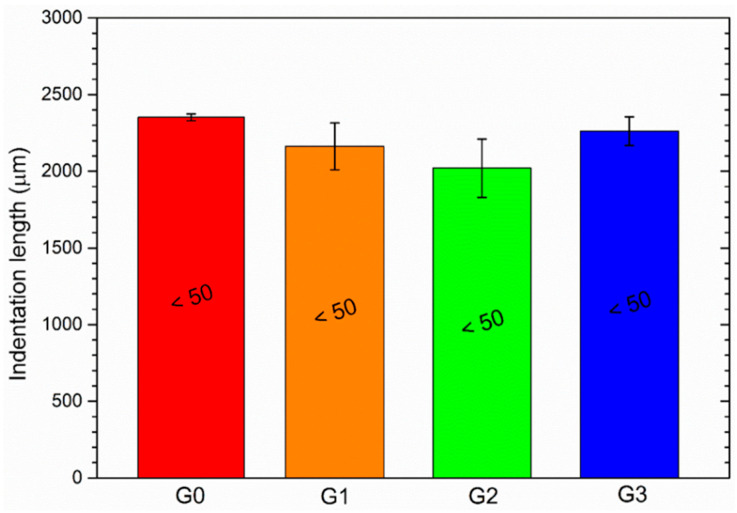
Indentation lengths of the Buchholz test notches, with the corresponding Buchholz hardness values.

**Figure 6 materials-16-02678-f006:**
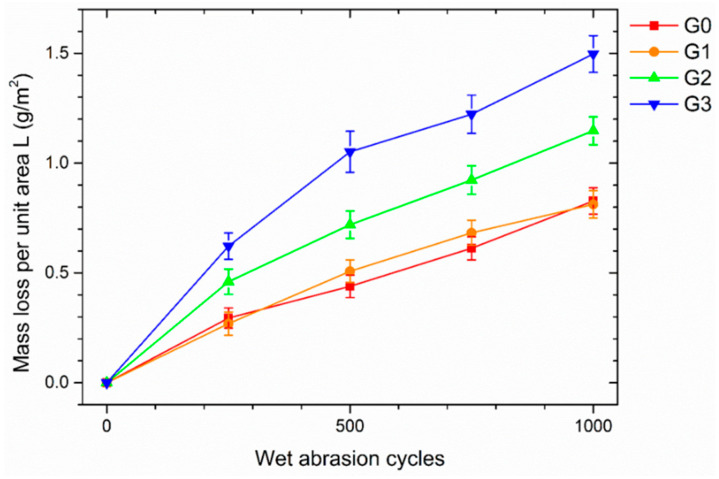
Loss in coatings mass per unit area, as a function of the abrasion cycles number.

**Figure 7 materials-16-02678-f007:**
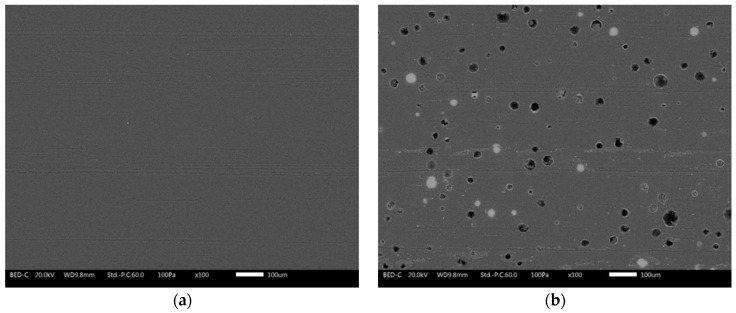
SEM micrographs of the surface morphology of (**a**) sample G0, (**b**) sample G1, (**c**) sample G2 and (**d**) sample G3 after the 1000 scrub test cycles (scale bar: 100 µm).

**Figure 8 materials-16-02678-f008:**
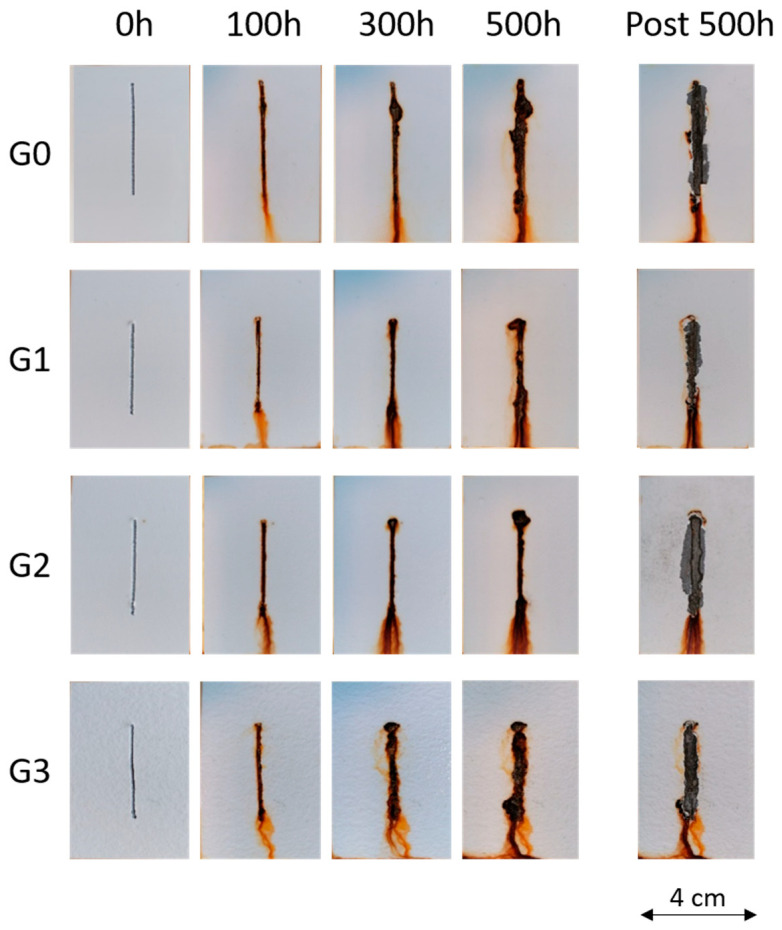
Coatings degradation during the exposure in the salt spray chamber.

**Figure 9 materials-16-02678-f009:**
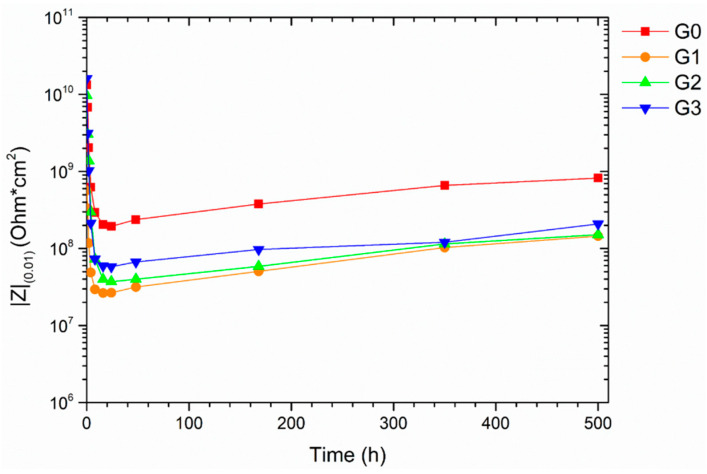
Bode impedance modulus |Z|_(0.01)_ evolution with time.

**Figure 10 materials-16-02678-f010:**
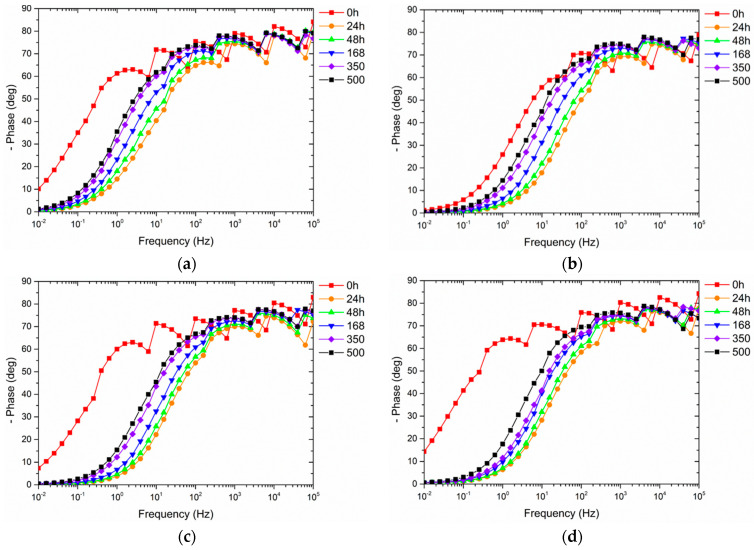
Bode phase spectra evolution with time of (**a**) sample G0, (**b**) sample G1, (**c**) sample G2 and (**d**) sample G3.

**Figure 11 materials-16-02678-f011:**
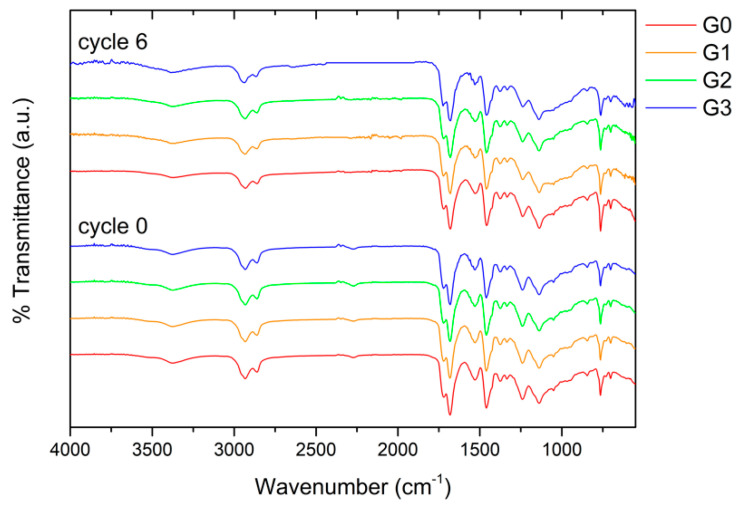
Evolution of the FTIR spectra of the samples before and after the 6 cycles of the accelerated degradation test.

**Figure 12 materials-16-02678-f012:**
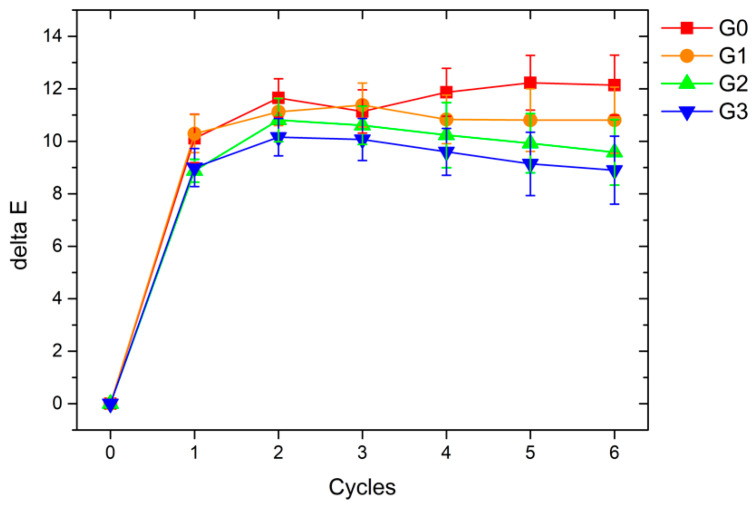
Color variation of the samples during the accelerated degradation test.

**Figure 13 materials-16-02678-f013:**
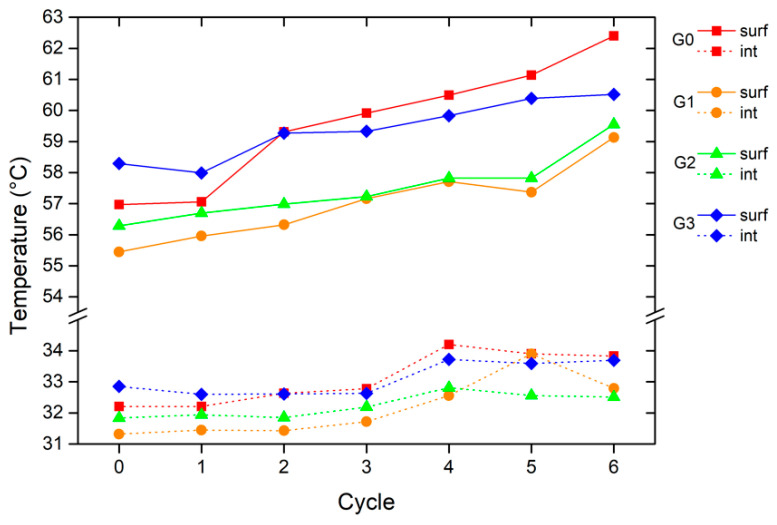
Evolution of the maximum measured T_surf_ and T_int_ as a function of the degradation cycles.

**Figure 14 materials-16-02678-f014:**
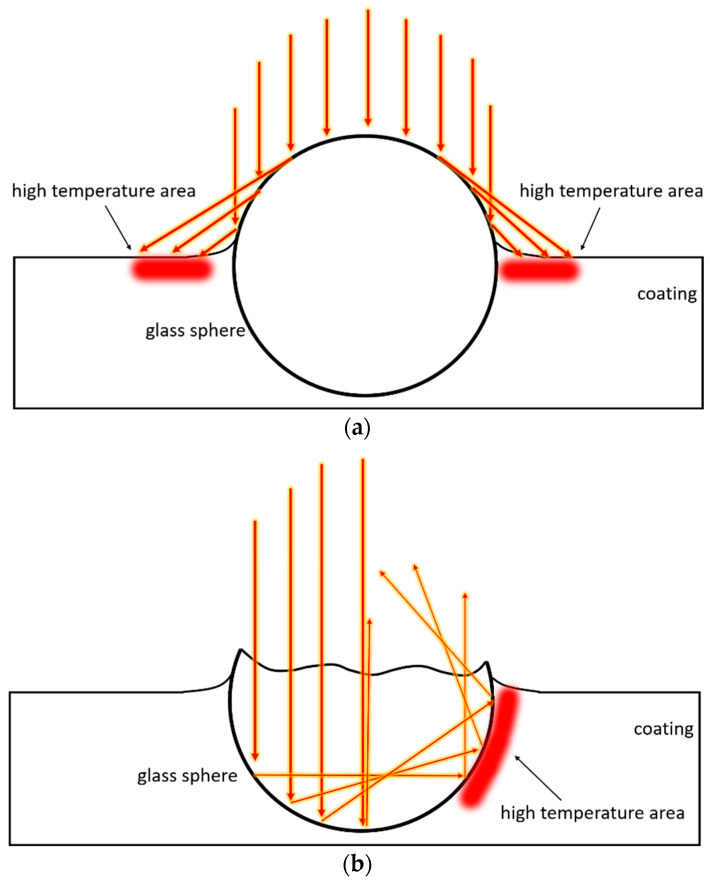
Scheme of local superheating processes, with (**a**) intact sphere and (**b**) broken sphere.

**Table 1 materials-16-02678-t001:** Samples nomenclature.

Samples Nomenclature	Microspheres in the Top-Coat [1 wt.%]
G0	/
G1	C40
G2	C15
G3	P

**Table 2 materials-16-02678-t002:** Coatings surface roughness.

Sample	Roughness Ra [µm]
G0	0.13 ± 0.01
G1	1.09 ± 0.06
G2	1.12 ± 0.02
G3	2.41 ± 0.12

**Table 3 materials-16-02678-t003:** Coatings detachment distance [µm].

Sample	Coating Detachment [µm]
G0	3692 ± 411
G1	3342 ± 672
G2	3742 ± 482
G3	3142 ± 724

## Data Availability

The data presented in this study are available on request from the corresponding author. The data are not publicly available due to the absence of an institutional repository.

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
