# Peer review of "Durability and Thermal Behavior of Functional Paints Formulated with Recycled-Glass Hollow Microspheres of Different Size"

_materials, 2023, doi:10.3390/ma16072678_

Round 1

Reviewer 1 Report

The EDS or XPS characterization is required in order to confirm the elements of the sample.

The meaning of "thermal behavior" in Fig.1 or section 2.3 is confused. What parameter is measured or what theretical model is used should be stated. 

Author Response

  1. The EDS or XPS characterization is required in order to confirm the elements of the sample.

Authors: The authors did not show the EDXS analyses of the samples, as the glass spheres, derived from recycled glass, are composed of approximately 70% SiO2, 15% Na2CO3 and 10% CaO. Once the spheres were introduced into the paint, as they were added in limited quantities, the analysed concentration of elements such as sodium and calcium was below the detector's credibility limit (< 0.5 wt.%). For these reasons, the authors decided not to include further information on the chemical composition of the samples (in a manuscript already very rich in graphs and data), as they are not very interesting and useful for the purpose of the work.

  1. The meaning of "thermal behavior" in Fig.1 or section 2.3 is confused. What parameter is measured or what theretical model is used should be stated.

Authors: with 'thermal behavior' the authors refer to the aspects of thermal conductivity of the coating and heat transmission within the house-model represented in Figure 1. The reference parameter is the temperature below the sample and inside the structure, analysed with the aid of thermocouples, as shown in Figure 1. The model of Figure 1 was used as already described and employed in previous literature works. The authors stated at line 154: “Finally, the impact of the microspheres on the thermal behavior of the paint was evaluated employing the experimental setup depicted in Figure 1 and already optimized in previous works [42,43].” At line 156 the authors added the following sentences: “The thermal behavior of the paint was evaluated considering the aspects of thermal conductivity of the coating and heat transmission within the house-model represented in Figure 1, measuring temperature changes in the system.”

Reviewer 2 Report

This study examined the possible usage of recycled-glass hollow microspheres in paints aimed for practical usage. All physical properties and performance tests were carried out which could prove to be valuable for industrial production of practical products. There are some comments and suggestions to be considered.

1. The introduction should contain some info on the literature review of the durability issue regarding composite paints.

2. What is the composition of the used recycled glass?

3. In Figure 3(a), there appeared no spherical shapes of the glass. Perhaps, other areas need to be examined. Also, the scale bars are not clearly readable. 

4. Line 246, "...good compatibility with the polymer..." was stated. Please explain more about this; would it be mechanical or chemical adhesion? What aspects would govern and judge such a compatibility? Not detecting any void or space at the interface does not necessarily imply good compatibility.

5. In Table 3, the standard deviations are quite large (400 - 700). How were they obtained? Line 358 stated that "...the results of the four samples are comparable." But with the shown error values, it appeared the G2 sample had the largest coating detachment. Any plausible reasons on such dependence on different sizes of microsphere?

6. Figure 9 shows that the sample G0 possessed a highest |Z| values at all frequencies. Addition of microspheres caused the samples to be more insulative. What would be a mechanism for this? Isn't glass very non-conductive?

7. Line 519: Please consider using the words "could lead" and "likely cause" as there was still no direct evidence empirically shown in this study regarding the two local overheating phenomena.

Author Response

  1. The introduction should contain some info on the literature review of the durability issue regarding composite paints.

Authors: As already written in the introduction, the mistake of many scholars is to evaluate only the multifunctionality of the filler, without investigating the effect of its additives on the durability of the paint. In fact, in the literature it is difficult to find works that also consider this aspect. However, the authors added the following sentence at line 73, adding some references: “In fact, in some cases the addition of filler leads to a reduction in the durability of the paint, due to inhomogeneity in the bulk of the layer [39], agglomeration phenomena [40], decay of the filler [41] or hydrophilicity features of the additive [42-43].”

  1. What is the composition of the used recycled glass?

Authors: the manufacturers of the glass spheres could not provide us with the exact composition. However, the glass spheres, derived from recycled glass, are composed of approximately 70% SiO2, 15% Na2CO3 and 10% CaO. The authors decide to avoid adding these information as they are not very functional for the purpose of the work, as the composition of the spheres is that of a standard glass. Once the spheres were introduced into the paint, as they were added in limited quantities, the analysed concentration of elements such as sodium and calcium was below the EDXS detector's credibility limit (< 0.5 wt.%). For these reasons, the authors decided not to include further information on the chemical composition of the samples (in a manuscript already very rich in graphs and data), as they are not very interesting and useful for the purpose of the work.

  1. In Figure 3(a), there appeared no spherical shapes of the glass. Perhaps, other areas need to be examined. Also, the scale bars are not clearly readable.

Authors: Figure 3(a) doesn’t reveal the glass spheres as it refers to sample G0, free of the glassy filler. Sample G0 represents the reference for all the analyses carried out in the study, as it is coated with the standard paint, free of fillers. The images in Figure 3 are representative of the internal structure of the coatings and their surface morphology. Since these features are constant in the coating, the authors didn't show other images (they would have appeared as unnecessary repetitions). The images have been modified with a higher scale bar.

  1. Line 246, "...good compatibility with the polymer..." was stated. Please explain more about this; would it be mechanical or chemical adhesion? What aspects would govern and judge such a compatibility? Not detecting any void or space at the interface does not necessarily imply good compatibility.

Authors: the good compatibility refers to the fact that the powders are not released following the brittle fracture process. Therefore, these are mechanical aspects rather than related to the chemical adhesion between glass and paint. Surely the absence of voids and spaces at the interface helps these compatibility aspects, even if they don't necessarily refer to good adhesion, as the reviewer suggests. The authors added the following sentence at line 234: “The compatibility between the fillers and paint matrix concerns physical and geometric aspects, rather than the adhesion between the polymer and glass. The absence of voids and spaces at the polymer-glass interface facilitates these compatibility issues, although they don't necessarily refer to good adhesion. The spherical glass filler is well received and surrounded by the polymer matrix, and the fracture of the coating does not result in the release of the entire sphere, but rather in its fracture into multiple components.”

  1. In Table 3, the standard deviations are quite large (400 - 700). How were they obtained? Line 358 stated that "...the results of the four samples are comparable." But with the shown error values, it appeared the G2 sample had the largest coating detachment. Any plausible reasons on such dependence on different sizes of microsphere?

Authors: Table 3 was obtained by evaluating the detachment of the coating, by means of optical microscope observations. Sample G2 has higher standard deviation, but also lower mean value. Considering both factors, it can be said that the four types of samples show similar behavior. In general, it is incorrect to define that adhesion changes as a function of the size of the spheres. These particularly high deviations are typical of this test, as the cathodic delamination is not homogeneous along the entire notch. The authors added the sentence at line 378: “The values were obtained by observing the samples with the optical microscope and measuring the length of the detachment of the coating from the central notch.”

  1. Figure 9 shows that the sample G0 possessed a highest |Z| values at all frequencies. Addition of microspheres caused the samples to be more insulative. What would be a mechanism for this? Isn't glass very non-conductive?

Authors: Impedance measurements can hardly highlight the insulating contribution of the filler. In fact, unexpectedly the |Z| values are lower in the samples containing the glass spheres, a notoriously insulating material. However, this phenomenon does not appear strange to the authors. Electrochemical impedance measurements are very sensitive analyses: a small defect in the coating, even a limited one, is enough to cause a collapse of the Z modulus. The glass spheres cannot cause an increase in the modulus of Z, as the electrons have a free path within the polymeric matrix. It is not possible to block the movement of electric charges if the spheres are not interconnected and do not create an effective barrier. Rather, the presence of the spheres causes the decrease of the modulus of Z as the filler introduces structural discontinuity in the bulk of the coating, which is 'felt' as a defect of the electrochemical system. Consequently, until the spheres reach a threshold value, they are not expected to cause the system impedance to increase. The authors added the following sentence at line 412: “Evidently, the filler causes the decrease of |Z|(0.01) as the filler introduces structural discontinuity in the bulk of the coating, which is 'felt' as a defect of the electrochemical system.”

  1. Line 519: Please consider using the words "could lead" and "likely cause" as there was still no direct evidence empirically shown in this study regarding the two local overheating phenomena.

Authors: the authors agree with the reviewer and have modified the sentence, as suggested.

Reviewer 3 Report

Nice and comprehensive study!

The paper analyzes different sizes hollow glass microspheres behaviour in insulating paint type coating. Beside thermal properties, mechanical resistance aspects were also studied.

The topic is original due to complex approach to the problem. Quite often such type of research is concentrated just to one aspect.

Other authors haven't treated insulating coating durability aspects.

There are no need to improve current research methodology.

Manuscript is written in solid format and conclusions are related with experimental evidences.

All references are proper and tables and figures correctly presented.

Author Response

Nice and comprehensive study!

The paper analyzes different sizes hollow glass microspheres behaviour in insulating paint type coating. Beside thermal properties, mechanical resistance aspects were also studied.

The topic is original due to complex approach to the problem. Quite often such type of research is concentrated just to one aspect.

Other authors haven't treated insulating coating durability aspects.

There are no need to improve current research methodology.

Manuscript is written in solid format and conclusions are related with experimental evidences.

All references are proper and tables and figures correctly presented.

Authors: the authors thank the reviewer for all the positive considerations.

Reviewer 4 Report

This study aims to assess the effect of hollow glass microspheres of different sizes derived from glass industry waste on the durability and thermal behavior of a waterborne paint. The results of this manuscript are interesting and it can be accepted after following revisions.

1- English of the manuscript needs polishing.

2- The main novelty of this work must be clearly mentioned as compared to other publications. 

3-More physical interpretation about the results can improve the quality of this work. 

4- What are the limitations of your experiments and research data?

5- It was mentioned that "After 1000 280 cycles, the mass loss of sample G3 is nearly double that of coating G0, free of filler.". What is the main reason behind this behavior?

Author Response

1- English of the manuscript needs polishing.

Authors: the authors have modified the structure of many sentences of the manuscript, improving the overall English level.

2- The main novelty of this work must be clearly mentioned as compared to other publications. 

Authors: the novelty of the work lies in evaluating the effect of the size of glass spheres, produced with recycled materials, as paint fillers for structural applications. Furthermore, a fundamental aspect of the study is the evaluation of the effect of the filler on the durability of the paint. As described in the Introduction, the works concerning the use of glass spheres as filler for paints do not consider this aspect. Thus, the authors have decided to add a sentence with some references at line 73, concerning the possible durability problems that arise when adding fillers in paints: “In fact, in some cases the addition of filler leads to a reduction in the durability of the paint, due to inhomogeneity in the bulk of the layer [39], agglomeration phenomena [40], decay of the filler [41] or hydrophilicity features of the additive [42-43].”

3-More physical interpretation about the results can improve the quality of this work. 

Authors: this comment is a bit vague, as it doesn't give useful indications to improve the work. The authors think they have already extensively described the test results, providing scientific explanations supported by literature references. However, the general text has been changed to provide more information.

4- What are the limitations of your experiments and research data?

Authors: The results shown in the manuscript are the result of several measurements carried out on multiple samples. Therefore, the authors can claim to have obtained good repeatability of the tests, just as the same type of sample always shows the same results. There are therefore no particular limitations either in the types of tests carried out, as regulated by very specific standards, or in the results obtained. The only non-standardized test is the one involving the set-up shown in Figure 1 for the measurements of the thermal aspects. However, this setup has already been used in several previous works and has always shown reliable and shareable results. The only limitation of the result, obviously, lies in the geometry of the house-model, which is not representative of a standard house. It is not possible to associate the temperature values with real measurable temperatures inside a real domestic structure, but the test setup allows to obtain a serious comparison between the performances of different samples.

5- It was mentioned that "After 1000 280 cycles, the mass loss of sample G3 is nearly double that of coating G0, free of filler.". What is the main reason behind this behavior?

Authors: The authors have already specifically described this phenomenon on line 312: “Consequently, the loss of mass measured during the Scrub test is associated both with the degradation of the polymer matrix and with the breakage of the filler, which is removed by the abrasive sponge. This phenomenon is more important the more the microspheres emerge from the bulk of the layer and are subject to shear stresses. Furthermore, for the same number of spheres destroyed, the loss of mass will be greater as a function of the size of the filler. Both two factors result in sample G3 showing the greatest mass loss during the test. The microspheres P are the largest (greatest mass) and consequently also those which are less protected by the coating and undergo greater shear stresses. The asperities shown in Figure 4 are more subject to abrasion phenomena, and therefore are also more easily removed from the abrasive sponge.
